# Chikungunya virus requires an intact microtubule network for efficient viral genome delivery

**Tabitha E. Hoornweg**[¤◉], **Ellen M. Bouma**[◉], **Denise P.I. van de Pol, Izabela A. Rodenhuis-Zybert, Jolanda M. Smit**[ID]*

Department of Medical Microbiology and Infection Prevention, University Medical Center Groningen, University of Groningen, The Netherlands

◉ These authors contributed equally to this work.
¤ Current address: Department of Biomolecular Health Sciences, Division of Infectious Diseases and Immunology, Faculty of Veterinary Medicine, Utrecht University, The Netherlands
* Jolanda.smit@umcg.nl

**Data Availability Statement:** All relevant data are within the manuscript and its Supporting Information files.

## Abstract

Chikungunya virus (CHIKV) is a re-emerging mosquito-borne alphavirus, which has rapidly spread around the globe thereby causing millions of infections. CHIKV is an enveloped virus belonging to the *Togaviridae* family and enters its host cell primarily via clathrin-mediated endocytosis. Upon internalization, the endocytic vesicle containing the virus particle moves through the cell and delivers the virus to early endosomes where membrane fusion is observed. Thereafter, the nucleocapsid dissociates and the viral RNA is translated into proteins. In this study, we examined the importance of the microtubule network during the early steps of infection and dissected the intracellular trafficking behavior of CHIKV particles during cell entry. We observed two distinct CHIKV intracellular trafficking patterns prior to membrane hemifusion. Whereas half of the CHIKV virions remained static during cell entry and fused in the cell periphery, the other half showed fast-directed microtubule-dependent movement prior to delivery to Rab5-positive early endosomes and predominantly fused in the perinuclear region of the cell. Disruption of the microtubule network reduced the number of infected cells. At these conditions, membrane hemifusion activity was not affected yet fusion was restricted to the cell periphery. Furthermore, follow-up experiments revealed that disruption of the microtubule network impairs the delivery of the viral genome to the cell cytosol. We therefore hypothesize that microtubules may direct the particle to a cellular location that is beneficial for establishing infection or aids in nucleocapsid uncoating.

## Author summary

Chikungunya virus (CHIKV) is an alphavirus that is transmitted to humans by infected mosquitoes. Disease symptoms can include fever, rash, myalgia, and long-lasting debilitating joint pains. Unfortunately, there is currently no licensed vaccine or antiviral treatment available to combat CHIKV. Understanding the virus:host interactions during the

**Funding:** T.E.H. was supported by an ASPASIA grant from the Dutch Scientific Organization (NWO) to J.M.S. E.M.B. was supported by the Graduate School of Medical Sciences of the University of Groningen and by a research grant from De Cock-Hadders Stichting of the University of Groningen. The funders had no role in study design, data collection and analysis, decision to publish, or preparation of the manuscript.

**Competing interests:** The authors have declared that no competing interests exist.

replication cycle of the virus is crucial for the development of effective antiviral therapies. In this study we elucidated the trafficking behavior of CHIKV particles early in infection. During cell entry, CHIKV virions require an intact microtubule network for efficient delivery of the viral genome into the host cell thereby increasing the chance to productively infect a cell.

## Introduction

Chikungunya virus (CHIKV) is a re-emerging mosquito-borne virus, which in the last decade caused millions of infections in the tropical and subtropical regions of the world [1]. CHIKV belongs to the alphavirus genus of the *Togaviridae* family. The CHIKV genome consists of a single positive-stranded RNA molecule of 11.8 kb in length, which is packaged by the CHIKV capsid (C) protein to form a nucleocapsid. The nucleocapsid is encapsulated by a host-cell derived lipid bilayer containing 240 copies of the CHIKV transmembrane glycoproteins E1 and E2. The envelope of the mature virus particle has an icosahedral T = 4 symmetry in which the E1 and E2 glycoproteins are arranged into 80 spikes, each spike comprising of 3 E1/E2-heterodimers (reviewed in [2]). Both E1 and E2 are important in the initial steps of infection. Whereas E2 facilitates binding to the host cell, E1 contains the hydrophobic fusion-loop and mediates membrane fusion [2].

Like most alphaviruses, CHIKV enters cells primarily via clathrin-mediated endocytosis (CME) [3–5]. Briefly, CHIKV binds to cell surface receptors that are localized or targeted to so-called clathrin-coated pits at the plasma membrane of cells. The clathrin-coated pit subsequently matures and will scission from the plasma membrane to form an intracellular clathrin-coated vesicle. Upon further internalization, the vesicle loses its clathrin coat and delivers the virus particle to the early endosome [6]. We previously showed that CHIKV co-localizes with clathrin for approximately 50 seconds, in accordance with the average life span of clathrin-coated pits/vesicles [5–7]. Membrane fusion is predominantly observed from Rab5-positive early endosomes [5]. CHIKV cell entry is a rapid process as 50% of the particles fused within 9 minutes post-addition of the virus to their host cells.

Intracellular transport of vesicles is highly regulated and primarily occurs on actin filaments and microtubules [8,9]. Of these, actin is mainly involved in movements close to the plasma membrane. These movements are either mediated by newly polymerizing actin filaments propelling an organelle or by myosin motor proteins trafficking along the actin filaments [9,10]. Additionally, newly formed endocytic vesicles are transported away from the plasma cortex in an actin polymerization-dependent manner [9,11]. During this transport, the vesicles often switch from an actin- to a microtubule-based movement. Microtubules generally provide long distance tracks for the transport of organelles to and from the perinuclear region [8,9]. The direction of the movement is governed by motor proteins traversing the microtubule. In most cell types dynein transports cargo towards the perinuclear region whereas kinesins direct their cargo towards the periphery [8,9].

The microtubule network has been implicated in the endocytic trafficking of multiple viruses, amongst others SV40 [12], HIV-1 [13], influenza A virus (IAV) [14,15], poliovirus [16], Ebola virus [17] and dengue virus [18]. For example, upon caveolae-mediated endocytosis, SV40-containing vesicles were found to move along microtubules towards the smooth ER. Microtubule disruption was found to block both the trafficking behavior as well as infection of SV40 [12]. HIV-1 was described to hijack microtubules for nuclear trafficking by interacting with adaptor proteins of dynein and kinesin [13,19]. Endocytic vesicles containing IAV are

targeted towards dynamic early endosomes in a microtubule-dependent manner [15]. Additionally, the endosomal trafficking of dengue virus is mediated by microtubules and disruption of microtubules was found to impair infection [18]. Retrograde polio virus trafficking inside neurons and Ebola virus cell entry and fusion were also found to be dependent on microtubules, yet the exact mechanism is not known [16,17]. Finally, Japanese encephalitis virus [20] and Crimean-Congo hemorrhagic fever virus [21] were described to require microtubules early in infection, yet the step at which microtubules function is unknown.

Here, we show that disruption of the cellular microtubule network impairs CHIKV infectivity at an early step in the replication cycle. Subsequent analysis of the intracellular trafficking behavior of CHIKV particles during cell entry revealed two distinct patterns; half of the particles showed fast-directed movement prior to arrival to early endosomes and membrane hemifusion whereas the other half of CHIKV particles did not show fast-directed movement. Disruption of the cellular microtubule network abolished the fast-directed movement of CHIKV particles during cell entry, influenced the localization of membrane hemifusion and impaired efficient viral genome delivery to the cell cytosol. Collectively, our data shows that trafficking along microtubules increases the chance to infection.

## Materials and methods

### Cells, inhibitors and plasmids

Green monkey kidney BS-C-1 cells (ATCC CCL-26) were cultured in Dulbecco's modified Eagle medium (DMEM) (Gibco, the Netherlands), high glucose supplemented with 10% fetal bovine serum (FBS) (Life Science Production, Barnet, United Kingdom), 25 mM HEPES, penicillin (100 U/ml), and streptomycin (100 U/ml) (Gibco). Human bone osteosarcoma epithelial U-2 OS cells (a gift from Mario Mauthe, University Medical Center Groningen, Groningen, The Netherlands) were maintained in DMEM, high glucose, glutaMAX supplemented with 10% FBS, penicillin (100 U/ml), and streptomycin (100 U/ml). Green monkey kidney Vero-WHO cells (ATCC CCL-81) were maintained in DMEM supplemented with 5% FBS, penicillin (100 U/ml), and streptomycin (100 U/ml). Baby hamster kidney cells (BHK-21 cells; ATCC CCL-10) were maintained in RPMI medium supplemented with 10% FBS, penicillin (100 U/ml), and streptomycin (100 U/ml). All cells were Mycoplasma negative and cultured at 37°C under 5% CO2.

Nocodazole was purchased from Sigma-Aldrich (St. Louis, Missouri, United States). The compound was dissolved in dimethyl sulfoxide (DMSO; BioAustralis, Australia) and stored according to the manufacturer's instructions. Bafilomycin A1 was diluted to a 200μM stock in DMSO. An ATPlite Luminescence Assay System (PerkinElmer, Waltham, Massachusetts, United States) was used to test for cytotoxicity, according to the manufacturer's instructions. Microtubule disruption upon nocodazole treatment was visualized by staining for α-tubulin. For this purpose, cells were treated with nocodazole after which they were fixed by 4% paraformaldehyde (PFA) (Alfa Aesar, Haverhill, Massachusetts, United States) and permeabilized using 0.1% Triton X-100 in PBS (Life Technologies, Carlsbad, California, United States). Subsequently, the samples were stained using mouse anti-α-tubulin (1:2000, Sigma) and rabbit anti-mouse AF647 (1:1000, Life Technologies).

The clathrin-LCa-eYFP plasmid was provided by X. Zhuang (Harvard University, Cambridge, MA, USA). The Rab5-wt-GFP plasmid was a gift from P. van der Sluijs (University Medical Center, Utrecht, The Netherlands).

### Virus production, purification, labeling and quantification

Virus production, purification and labeling procedures have been described previously [5]. Briefly, CHIKV strains LR2006-OPY1, LS3 and LS3-GFP [22] were produced by

electroporation of *in vitro*-transcribed RNA transcripts into BHK-21 cells. Virus working stocks were produced by subsequent passage in Vero-WHO cells.

For large scale production, BHK-21 monolayers were inoculated with CHIKV at an MOI of 4. At 24 h post-infection (hpi) the supernatant was harvested and cleared from cell debris by low-speed centrifugation. Subsequently, the virus was pelleted by ultracentrifugation, resuspended in HNE and purified by ultracentrifugation on a sucrose gradient (20 to 50% [wt/vol] in HNE). The 40%-to-45% section containing the virus was harvested, aliquoted, and stored at −80˚C.

The infectious virus titer in plaque forming units (PFU) was determined by a standard plaque assay using Vero-WHO cells. Plaques were counted 2 days after infection. The number of genome equivalent particles (GCPs) were determined by reverse transcriptase PCR (RT-PCR) followed by quantitative PCR (qPCR), as described previously [23].

For live cell microscopy assays, CHIKV was labeled with the lipophilic fluorescent probe DiD (1,1′-dioctadecyl-3,3,3′,3′-tetramethylindodicarbocyanine, 4-chlorobenzenesulfonate salt; Life Technologies) at experimental conditions that do not interfere with virus infectivity, as described previously [5]. DiD-labeled virus was stored at 4˚C in the dark and was used within 2 days.

## Single-particle tracking of DiD-labeled CHIKV

Single-particle tracking experiments were performed as described previously [5]. Briefly, BS-C-1 cells were seeded into Nunc 8-well Lab-Tek II chambered coverglass slides (Thermo Scientific) to obtain 50 to 70% confluence on the day of tracking. When indicated, $1.25 \times 10^6$ BS-C-1 cells were transfected with 5 µg of plasmid DNA by electroporation before seeding.

Cells were washed three times with phenol red-free MEM (Gibco). Subsequently, phenol red-free MEM supplemented with 1% glucose (Merck, Darmstadt, Germany) was added to the cells in presence or absence of 10 µM nocodazole. When nocodazole was added, cells were pre-incubated for 2 h to allow disruption of the microtubules. Just before microscopy, GLOX was added to prevent phototoxicity [24]. Cells were mounted on the Leica Biosystems 6000B microscope and kept at 37˚C throughout the whole experiment. DiD-labeled CHIKV was added *in situ*, and image series were recorded at 1 frame per s for 25 to 30 min. To localize the nucleus and plasma membrane of the cell, DIC snapshots were taken before and after the imaging. The location of fusion was referred to as 'perinuclear' when the particle fused in an area that was closer to the nucleus than the plasma membrane.

Image processing and analysis were carried out using ImageJ (NIH) and Imaris x64, release 7.6.1 (Bitplane Scientific Software). Trajectories were generated using the 'particle tracking' function of Imaris by pairing peaks per frame to previously established trajectories according to proximity and similarity of intensity. For the generated trajectories, the 'particle tracking' function of Imaris calculated the particle velocity. The fluorescence intensity during a trajectory was quantified using an *in-house* macro based on the 'particle analyzer' plugin of ImageJ. A sudden (within 1 to 2 seconds) >2-fold increase in fluorescence intensity was defined as the moment of membrane fusion. As we cannot distinguish between hemifusion and complete membrane fusion in this assay we specifically refer to hemifusion. Membrane hemifusion is a transient state prior to complete membrane fusion in which the outer leaflets of the fusing membranes have fused (thereby allowing DiD redistribution) but the inner leaflets are still intact. Only particles with a fluorescent level of $\leq 40$ arbitrary units (a.u.) were considered individual particles and were selected for analysis. Additionally, as the thickness of the cell close to the nucleus is larger than one focal plane [5], particles that bound to the cell close to the nucleus were excluded from further analysis. Upon simultaneous tracking of the virus and either clathrin or Rab5, co-localization was assessed by eye.

## Microscopic fusion assay

The membrane hemifusion capacity of CHIKV in the presence and absence of 10 μM nocodazole was estimated by a microscopy-based fusion assay. For this purpose, BS-C-1 cells were seeded into Nunc 8-well Lab-Tek II chambered coverglass slides to obtain a sub-confluent monolayer the next day. Cells were washed three times with serum-free, phenol red-free MEM. Subsequently, phenol red-free MEM containing 1% glucose was added in presence or absence of nocodazole and cells were incubated for 2 h at 37˚C. Next, DiD-labeled CHIKV was added to the cells at MOI 20 after which cells were incubated at 37˚C for another 30 min in order to allow viral fusion. Unbound virus was removed by washing three times with serum-free, phenol red-free MEM after which fresh phenol red-free MEM containing 1% glucose was added.

Microscopic analysis was done with the Leica Biosystems 6000B instrument. Fields were randomly selected in the differential interference contrast (DIC) settings, after which snapshots were taken in both the DIC and DiD channel. Per experiment a total of 20 random snapshots were analyzed using the 'particle analyzer' plugin of ImageJ. The total area of fluorescent spots was quantified in a.u. for each snapshot and averaged per experiment.

## Flow cytometry analysis of infection

Flow cytometry analysis was used to assess the effects of nocodazole on CHIKV infection. For this purpose, 10 μM nocodazole was diluted in cell culture medium containing 2% FBS. Cells were pre-incubated with nocodazole for 2 h, after which CHIKV was added to the cells at the indicated MOIs. At 1.5 hpi, fresh cell culture medium containing 10% FBS and 10 μM nocodazole was added. At 10 hpi, cells were washed with PBS, trypsinized and fixed with 4% PFA. The cells infected with CHIKV strain LR2006-OPY1 were additionally permeabilized and stained with a rabbit anti-E2-stem antibody (1:1000; obtained from G. Pijlman, Wageningen University, Wageningen, The Netherlands) and Alexa Fluor 647-conjugated chicken anti-rabbit antibody (1:300; life Technologies). Subsequently, cells were analyzed by flow cytometry (roughly 30.000 cells per condition).

For the time-chase experiments, BS-C-1 cells were treated with 10 μM nocodazole diluted in BS-C-1 medium for either for the complete (-2–10 hpi), at an early (-2–0.5 hpi) or late (0.5–10 hpi) stage of infection. CHIKV LS3-GFP was added at an MOI of 20. At 0.5 hpi the virus inoculum was removed and cells were washed thoroughly to remove unbound particles. Subsequently, fresh BS-C-1 medium containing 10% FBS was added in the presence or absence of 10 μM nocodazole. Infection was continued until 10 hpi, after which cells were prepared for flow cytometry as described above.

## Cell fractionation

Virus genome delivery was estimated by use of a cell fractionation protocol followed by PCR-based quantification of the viral genomic (+)-strand RNA. U-2 OS cells were pretreated for 2 h with 10 μM nocodazole, or 1 h with 1 μM Bafilomycin A1 in U-2 OS medium containing 2% FBS, after which they were infected with CHIKV LR2006-OPY1 for 1.5 h at an MOI of 5. Subsequently, cells were washed thoroughly to remove unbound virus. In order to fractionate the cells, cells were permeabilized by incubation with 50μg/mL digitonin (Sigma-Aldrich) for 5 min at RT and subsequently for 30 min on ice [25]. Afterwards, the supernatant was carefully collected to obtain the cytosolic fraction and the remaining permeabilized cells were directly lysed in the cell culture plate to obtain the membrane fraction. Subsequent RNA isolation was performed using the viral RNA kit and the RNAeasy mini kit for the cytosolic and the membrane fraction, respectively, according to manufacturer's instructions. Subsequently, samples

were subjected to RT-qPCR, as described before [23]. After cDNA synthesis, the samples were treated with 10 units of RNAse A (Thermo Scientific) for 1 h at 37˚C to degrade viral RNA. To control for efficient cell fractionation, the protein levels for the cytosolic marker GAPDH and the vesicular marker Rab5 were assessed by Western blot. For this purpose, fractionated samples were loaded onto precast 10% Mini-PROTEAN TGX gels (BioRad). Gels were blotted using the Trans-Blot Turbo Transfer system (Biorad) and Trans-Blot Turbo Mini PVDF Transfer Packs (Biorad). Mouse anti-GAPDH (1:10000; Abcam, Cambridge, United Kingdom) and rabbit anti-Rab5 (1:1000; Abcam) were used as primary antibodies. Secondary HRP-conjugated antibodies (Thermo Fisher Scientific) were used as recommended by manufacturer.

## Statistical analysis

All data were analyzed in GraphPad Prism software. Data are presented as mean ±SD. Student T test was used to evaluate statistical differences. P value $\leq 0.05$ was considered significant with $^{*}p \leq 0.05$, $^{**}p \leq 0.01$, $^{***}p \leq 0.001$ and $^{****}p \leq 0.0001$.

## Results

### An intact microtubule network is important early in CHIKV infection

To study whether microtubules play a role in CHIKV infection, we first determined the effect of the microtubule-depolymerizing agent nocodazole on CHIKV infectivity in human bone osteosarcoma epithelial (U-2 OS) cells as epithelial cells are considered natural target cells during CHIKV infection [2,26]. Nocodazole was added to the cells 2 h prior to infection and remained present throughout the duration of the experiment at an end-concentration of 10 μM. Incubation of the cells for 2 h with 10 μM nocodazole was sufficient to disrupt the microtubule network (S1A Fig). Furthermore, 10 μM nocodazole was well-tolerated by the cells for the duration of the experiment (S1B Fig). Cells were infected with the clinical isolate CHIKV LR2006-OPY1 and the synthetic CHIKV strain LS3-GFP [22] at a multiplicity of infection (MOI) of 1 and 20. The amino acid sequence of the structural proteins is identical between CHIKV LR2006-OPY1 and CHIKV LS3-GFP, yet LS3-GFP has the advantage of read-out on the basis of GFP expression. The number of CHIKV-infected cells was determined at 10 hpi, which reflects 1 round of replication [27]. In the absence of nocodazole, CHIKV LR2006-OPY1 infected 2.9% ± 1.5% and 25.3% ± 12.7% of the cells at MOI 1 and 20, respectively (S2A Fig). In case of CHIKV LS3-GFP, 3.8% ± 0.1% and 35% ± 1.1% of the cells were infected in the absence of nocodazole at MOI 1 and MOI 20, respectively (S2B Fig). Importantly, upon treatment of the cells with nocodazole, the number of infected cells were 70% reduced following infection with both viruses and at both MOIs when compared to the control (Fig 1A). Only minor differences were observed in the mean fluorescent intensity (MFI) of the CHIKV-infected cell population in the absence and presence of nocodazole, suggesting that once a cell is infected the translation/replication processes are not affected (Fig 1B). Since CHIKV LR2006-OPY1 and CHIKV LS3-GFP exhibit similar properties these viruses will be used intertwined in this study.

Furthermore, a reduction in infectious virus particle production from 5.2 ± 0.1 Log PFU/ mL in control cells to 4.5 ± 0.7 Log PFU/mL in nocodazole-treated cells was seen following infection with CHIKV LR2006-OPY1 at MOI 1. At MOI 20, a reduction from 6.5 ± 0.6 Log PFU/mL to 5.7 ± 0.3 Log PFU/mL was seen upon treatment of the cells with nocodazole (Fig 1C). Thus, on average 80% reduction in the secretion of infectious virions is seen following infection of nocodazole-treated cells, which is in line with the observed decrease in the number of infected cells (Fig 1A).

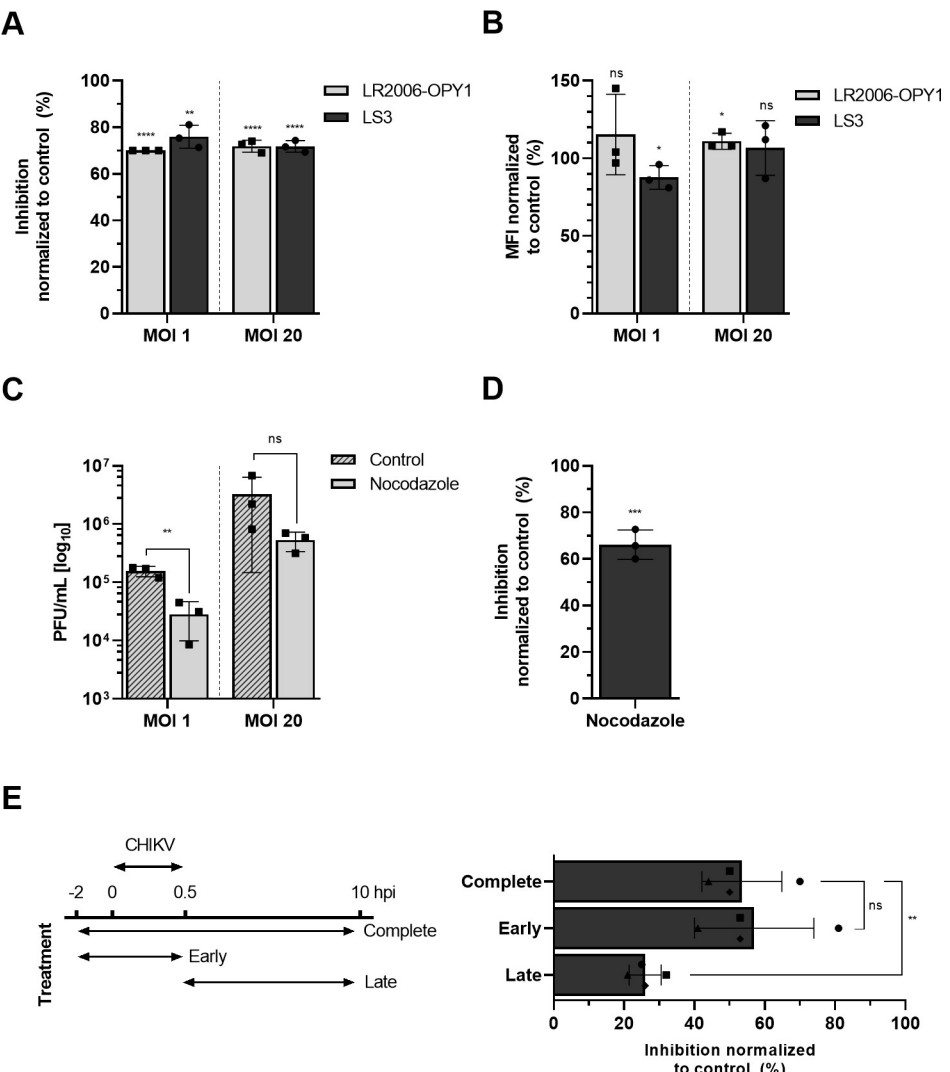

**Fig 1. Microtubules are required in the early steps of CHIKV infection.** U-2 OS cells were pretreated for 2 h with 10 μM nocodazole and infected with CHIKV LS3-GFP (A, B) or CHIKV LR2006-OPY1 (A, B, C) at MOI 1 or 20 for 10 h. (A) Flow cytometry analysis of CHIKV infection in presence of nocodazole. The percentage of inhibition is normalized to the non-treated control. (B) Mean fluorescence intensity (MFI) of the infected population is normalized to the non-treated control. (C) Supernatants were harvested and virus particle production was analyzed by plaque assay on Vero-WHO cells. (D) Flow cytometry analysis of CHIKV infection in BS-C-1 cells. The cells were pretreated for 2 h with 10 μM nocodazole and infected with CHIKV LS3-GFP at MOI 20. The percentage of inhibition is normalized to the non-treated control. (E) Flow cytometry-based time-of-addition assay of CHIKV infection in BS-C-1 cells. Nocodazole (10 μM) was added to the cells during early (-2–0.5 hpi), late (0.5 h– 10 hpi) or complete (-2–10 hpi) time of infection. CHIKV LS3-GFP (MOI 20) was removed at 0.5 hpi, after which cells were washed and incubation was continued for another 9.5 h. The percentage of inhibition is normalized to the non-treated control. Data represents at least three independent experiments each performed in triplicate, each dot represents the average of an individual experiment; error bar represents SD. Data was analyzed and compared to the control using student T-test ($^*$p $\leq$ 0.05, $^{**}$p $\leq$ 0.01, $^{***}$p $\leq$ 0.001 and $^{****}$p$\leq$ 0.0001). ns, not significant.

To further confirm that an intact microtubule network is important for CHIKV infectivity, we next evaluated the effect of nocodazole on CHIKV infectivity in the kidney epithelial cell line BS-C-1. Similar as with U-2 OS cells, 2 h incubation with 10 μM nocodazole disrupted the microtubule network in BS-C-1 cells (S3A Fig). Furthermore, nocodazole treatment was tolerated up to at least 12 h of incubation (S3B Fig). Cells were infected with CHIKV LS3-GFP at

MOI 20 and the number of infected cells was determined at 10 hpi. Importantly, the number of CHIKV-infected BS-C-1 cells decreased from 9.2% ± 4.7% in control cells to 3.2% ± 0.9% in nocodazole-treated cells (S4A Fig), resulting in an inhibition in infection of approximately 65% (Fig 1D). The MFI of the infected cell population was 28.8% ± 12% reduced (S4B Fig). Taken together, the above results clearly show that an intact microtubule network promotes CHIKV infectivity.

Next, time-of-addition experiments were performed to assess at which stage of the replicative cycle CHIKV requires an intact microtubule network. Briefly, BS-C-1 cells were treated with nocodazole at an 'early' (2 h prior to infection to 0.5 hpi) and 'late' (0.5 hpi to 10 hpi) stage of infection or during the complete experiment (2 h prior to infection to 10 hpi). At 0.5 hpi, unbound CHIKV particles were removed by thoroughly washing the cells. After washing, cell culture medium was added for the remaining 9.5 h of the experiment. In the 'late' and 'complete' conditions, nocodazole was added to the medium. We observed 54% ± 11% inhibition of infection when cells were treated with nocodazole for the complete duration of the experiment (Fig 1E). This is slightly lower than in Fig 1D but this is likely due to differences in the experimental set-up of these experiments. Interestingly, a comparable inhibition, 57% ± 17%, was seen when nocodazole was present during the early stages of infection (-2 h– 0.5 hpi). Addition of nocodazole after 0.5 hpi perturbed CHIKV infection by only 26% ± 4.5%. These findings show that microtubules are predominantly required during the early steps of CHIKV infection.

## CHIKV exhibits two types of trafficking behavior upon endocytosis

The observation that microtubule disruption predominantly has an effect in the first 0.5 h of infection is suggestive of their importance during CHIKV cell entry. To study this in more detail, we examined the intracellular trafficking behavior of CHIKV particles prior to fusion by live-cell imaging of DiD-labeled CHIKV particles, as described before [5]. Briefly, the lipophilic DiD probe was incorporated into the CHIKV membrane to such an extent that the fluorescence was largely quenched [5]. At these labeling conditions, membrane hemifusion can be observed as a sudden increase in fluorescence intensity due to dilution of the probe in the target membrane (5). Single particle tracking of DiD-labeled CHIKV was performed in BS-C-1 cells as the relative flatness of these cells increase the chance of capturing complete viral trajectories, thus from virus-cell binding to viral hemifusion, in the epi-fluorescence microscopy setup applied.

In order to track entry of CHIKV virions into BS-C-1 cells, DiD-labeled CHIKV particles were added to the cells *in situ* and images were recorded at 1 frame per second for a total of 25 min. CHIKV exhibited two different types of trafficking behavior during cell entry (Fig 2A and 2B). In the first type of trafficking, virions remained relatively immobile at first, after which they suddenly exhibited a fast and directed movement. Thereafter, the virus moved relatively slowly and viral hemifusion was seen (Fig 2A left panels). The fast-directed movement is visible as a clear increase in velocity when particle velocity is plotted against the time (Fig 2A right panels). A movie showing this virus trafficking behavior is added as supplementary data (S1 Movie). Other virus particles remained relatively static till the moment of membrane hemifusion (Fig 2B left panels). Consequently, considerably shorter distances were covered by the virions. As expected, when viral velocity was plotted against time no distinct peaks were observed (Fig 2B right panels). S2 Movie shows an example of this type of trafficking behavior.

In total 29 CHIKV trajectories were recorded to assess the frequency of the distinct types of trafficking behavior and the time to membrane hemifusion. Fourteen CHIKV particles (48%) exhibited fast-directed movement before hemifusion whereas the other 52% remained static

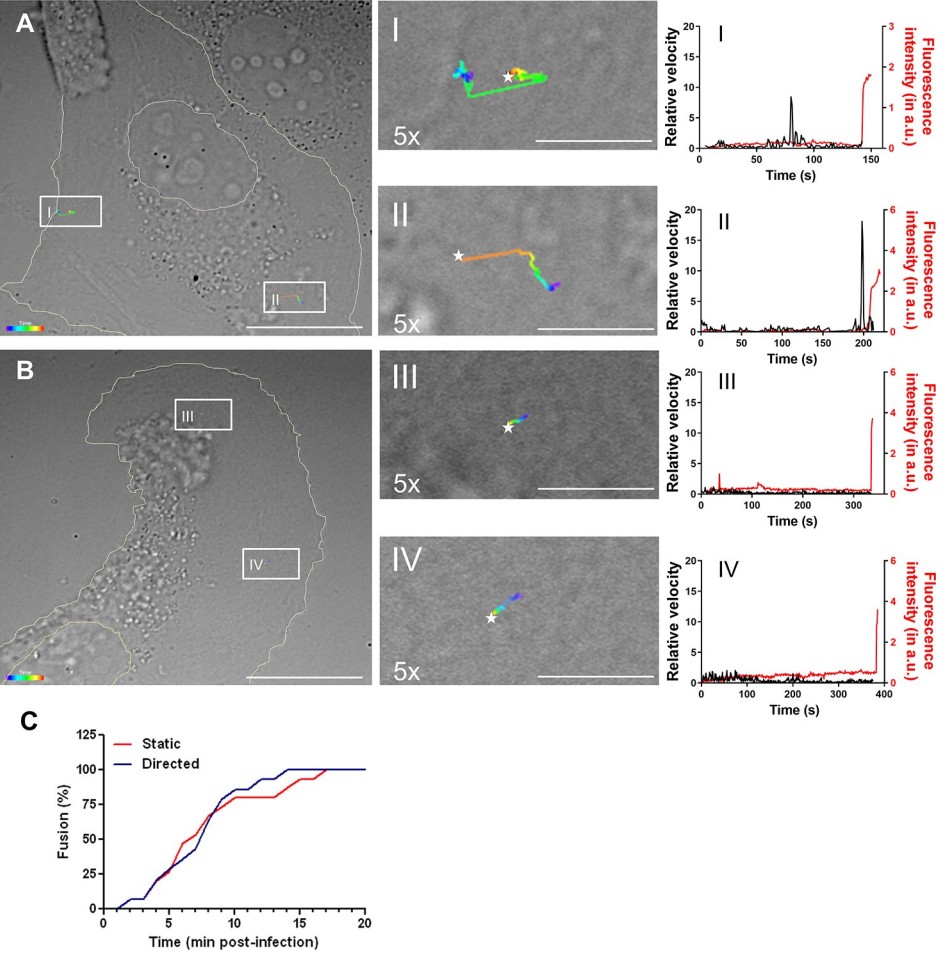

**Fig 2. Trajectory analysis of CHIKV cell entry.** (A) Two representative single virus particles exhibiting fast-directed movement before hemifusion. (B) Two representative particles that remain quite static during cell entry. Left panel (A and B) shows images of a cell obtained with DIC optics. On top of the DIC image the trajectories of two single virus particles are projected. Trajectories are depicted color-coded, with purple representing the start and red representing the end of the trajectory. Hemifusion is indicated by a white star. Scale bar represents 25 μm. In the middle panels the trajectories are 5 times enlarged (scale bar represents 5 μm) for visual purposes. Right panel shows the relative velocity (in black) and fluorescence intensity (in red) of a single virus particle over time. A sudden increase in fluorescence intensity indicates the moment of hemifusion. The data correspond to the trajectories depicted in the left and middle panels of A and B. (C) Analysis of the time to hemifusion. In this analysis, the hemifusion time point of 14 particles exhibiting directed movements and 15 particles that remained static was used.

during cell entry. These results indicate that fast-directed and static trafficking behaviors are equally common. Furthermore, no substantial differences in the time to membrane hemifusion were observed (Fig 2C). Half of the virus particles that exhibited fast-directed movement fused within 8 min post-infection (mpi) and half of the static particles fused within 7 mpi (Fig 2C).

## Microtubules are required for CHIKV fast-directed movement and hemifusion in the perinuclear region of the cell

To investigate whether the fast-directed movement resembles transport along microtubules, we next performed tracking experiments in cells pre-treated with 10 μM of nocodazole. Fig 3A shows two representative examples of CHIKV trajectories in nocodazole-treated cells. In both

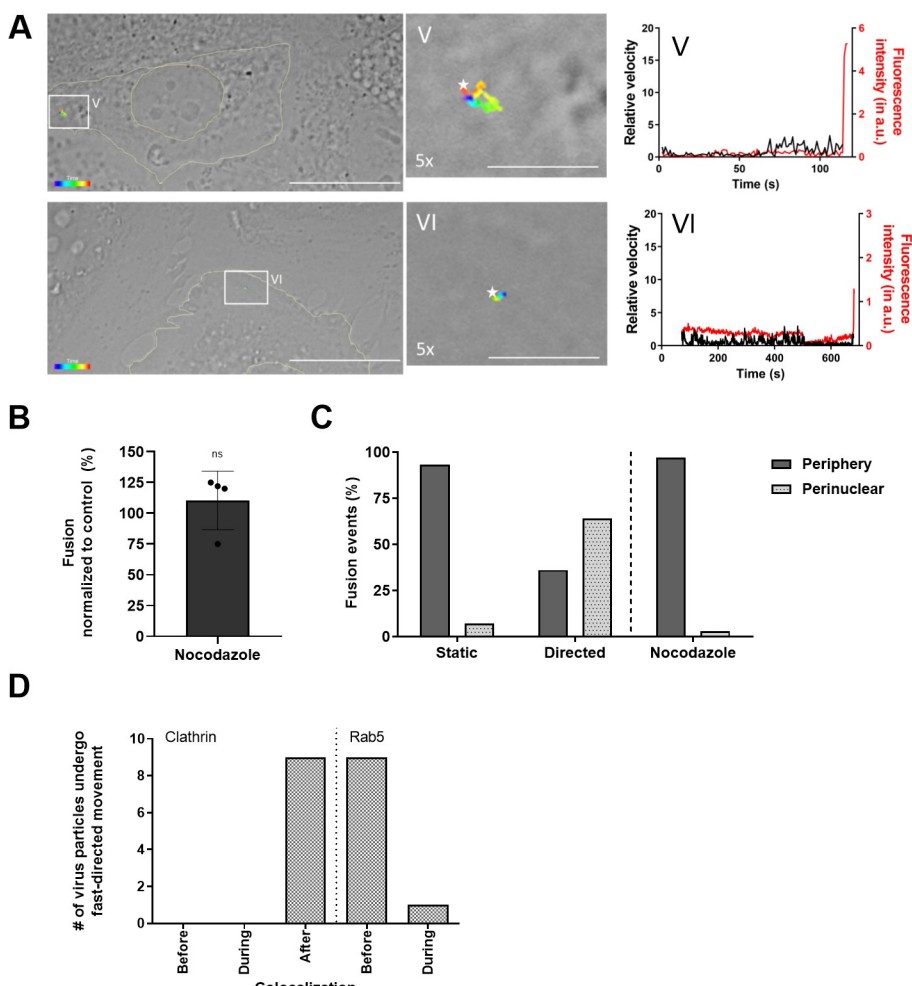

**Fig 3. Fast-directed movements by microtubule regulate the location, but not the amount of membrane hemifusion.** (A) Representative CHIKV trajectories in nocodazole-treated cells, similar as described in the legend to Fig 2A and 2B. (B) Microscopic fusion assay in BS-C-1 cells pretreated with 10 μM nocodazole, as described in Materials and Methods. The graph represents four independent experiments each performed in duplicate. Data is normalized to non-treated control. Each dot represents the average of an individual experiment. Data was analyzed and compared to the control using student T-test (ns, not significant). Error bars represent SD. (C) Analysis of the location of hemifusion. In control cells, 15 'static' particles and 14 particles showing fast-directed movement were analyzed. In nocodazole-treated cells, 34 particles were analyzed. (D) BS-C-1 cells were transfected with clathrin-YFP (left panel) or Rab5-GFP (right panel) and subjected to dual-color single particle tracking using DiD-labelled CHIKV. A total of 9 and 10 virus trajectories showing fast-directed movement in BS-C-1 cells transiently expressing clathrin-YFP or Rab5-GFP, respectively, were used to determine the moment of fast-directed movement as before, during or after colocalization with clathrin-YFP or Rab5-GFP.

trajectories no fast-directed movements were observed. Furthermore, when plotting the velocity of the virus particle over time no sudden peaks in velocity were apparent (Fig 3A right panels). In total, 36 CHIKV trajectories were recorded in nocodazole-treated cells and fast-directed movement was observed in only 3 (8%) trajectories, indicating that the fast-directed movement indeed resembles trafficking along microtubules. Representative movies are provided in S3 Movie and S4 Movie. Interestingly, although nocodazole treatment impaired CHIKV fast-directed movement, membrane hemifusion could still be observed (example given in Fig 3A). To quantitatively assess whether microtubule disruption affects CHIKV cell entry and hemifusion, we next employed a microscopic fusion assay [5,24]. In this assay DiD-

labeled CHIKV is incubated with BS-C-1 cells for 30 min. Subsequently, unbound virus is washed away and 20 random images are taken using both DIC and DiD optics. The total extent of DiD fluorescence is taken as a measure of CHIKV cell entry and hemifusion. In correspondence to our single particle tracking results, no differences were observed between cells treated with and without nocodazole, indicating that nocodazole-treatment does not impair the extent of CHIKV cell entry and hemifusion (Fig 3B).

We next evaluated the location of CHIKV hemifusion in the cell. Under normal infection conditions, the vast majority of the static particles fuse in the periphery, whereas more than half of the particles that are transported via fast-directed movement fuse close to the perinuclear region (Fig 3C). Upon nocodazole treatment, the vast majority of particles fused in the cell periphery (Fig 3C). Thus, microtubule-dependent movement is a prerequisite for CHIKV hemifusion in the perinuclear region of the cell.

## CHIKV fast-directed movement occurs after clathrin-mediated internalization yet before delivery to the early endosome

We and others previously published that CHIKV mainly enters mammalian cells via CME [3–5]. After internalization, the viral particles are delivered to Rab5-positive early endosomes, from which CHIKV fusion primarily occurs [5,28]. In order to assess at what stage of internalization the microtubule-dependent movement occurs, we transfected BS-C-1 cells with clathrin-YFP and performed dual-colour single particle tracking using DiD-labeled CHIKV. For this analysis, 9 CHIKV trajectories were used that displayed fast-directed movement prior to hemifusion. In all trajectories, fast-directed movement was seen after co-localization of CHIKV with clathrin, suggesting that microtubule-dependent trafficking occurs post-CME (Fig 3D, left panel). We previously showed that posterior to CME, CHIKV resides in a clathrin- and Rab5-negative endocytic vesicle for on average 14 s [5]. Thereafter, the virus is delivered to the Rab5-positive early endosome, where it resides for on average 38 s till membrane hemifusion occurs (5). Therefore, we next zoomed into the endocytic stages post-CME and traced DiD-labeled CHIKV in BS-C-1 cells transfected with the early endosomal marker Rab5-GFP. In 9 out of 10 trajectories analysed, fast-directed movement occurred just before the co-localization between CHIKV and Rab5. In 1 trajectory, fast-directed movement was observed when the virus was already delivered to a Rab5-positive early endosome (Fig 3D, right panel). These results demonstrate that CHIKV fast-directed movement occurs after internalization, but before delivery of the virus to the early endosome. Consequently, during fast-directed movement, CHIKV likely resides in a clathrin- and Rab5-negative endocytic vesicle.

## Nocodazole impairs CHIKV genome delivery

The above data clearly indicates that membrane hemifusion occurs irrespective of microtubule-dependent movement. Hemifusion is a transient intermediate in the fusion process that precedes the formation of a fusion pore and subsequent delivery of the nucleocapsid/viral RNA to the cell cytosol. To assess whether microtubules are important for successful nucleocapsid/genome delivery, we next employed a cell fractionation protocol followed by the quantification of the CHIKV genomic RNA (gRNA) within the membrane (non-fused) and cytosolic (fused) fraction using qPCR. In these experiments, U-2 OS were used as these cells are slightly more permissive to infection than BS-C-1 cells. Cells were fractionated by use of digitonin treatment, which specifically permeabilizes the plasma membrane [25,29]. First, we checked the fractionation efficiency in four independent experiments by Western blot. A representative blot is shown in Fig 4A. Quantification of the protein bands revealed that the vast majority of the vesicular marker Rab5 (73% ± 11%) and the cytosolic marker GAPDH (86% ±

 

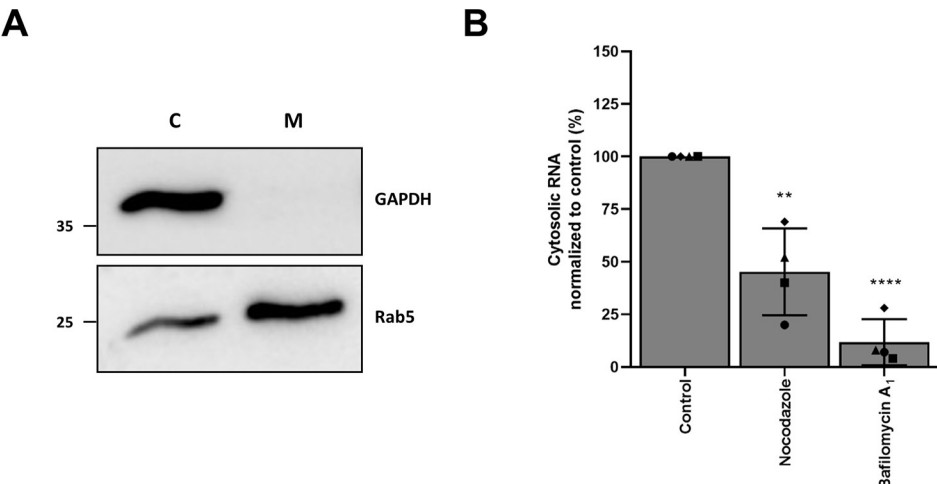

**Fig 4. Microtubule disruption impairs viral genome delivery.** Detection of CHIKV gRNA in membrane and cytosolic fractions by RT-qPCR. (A) Representative Western blot. Shown is cytosolic (C) and membrane (M) fraction. (B) Total number of CHIKV gRNA was assessed by RT-qPCR. Percentage of cytosolic gRNA normalized to non-treated control is shown. Four independent experiments were performed in duplicate. Each dot represents the average of an independent experiment. Error bars represent SD. Data was analyzed and compared to the control using student T-test (**p ≤ 0.01 and ****p ≤ 0.0001).

7.4%) were found in the anticipated cellular fractions (S5 Fig). Treatment of cells with the vacuolar H$^+$-ATPase inhibitor bafilomycin A1 prevents endosome acidification thereby inhibiting CHIKV membrane fusion and was used as a positive control in the assay. Fig 4B shows that there is a 2-fold decrease (55% ± 21%) in the quantity of cytosolic CHIKV gRNA in nocodazole-treated cells when compared to that of the non-treated control. As expected, bafilomycin A1 treatment inhibited genome delivery to the cytosol with 88% ± 11% when compared to the non-treated control. Taken together, the above data shows that microtubules aid in the successful delivery of the gRNA into the cytosol.

## Discussion

We observed that CHIKV particles have distinct patterns of intracellular trafficking prior to inducing membrane hemifusion. Half of the CHIKV particles exhibit 'static' behavior and predominantly fuse in the cell periphery whereas the other half traffic along microtubules during cell entry and predominantly fuse in the perinuclear region of the cell. Microtubule-mediated trafficking occurs after clathrin-mediated endocytosis but prior to delivery of the particle to the early endosome. Disruption of the microtubule network reduced the number of infected cells yet did not influence the extent of membrane hemifusion activity. At these conditions, hemifusion was predominantly seen in the cell periphery and CHIKV gRNA delivery to the cell cytosol was impaired. In summary, this study sheds light on the intracellular trafficking behavior of CHIKV and demonstrates that microtubules play an important role during the early steps of CHIKV infection.

A possible explanation to our findings is that trafficking along microtubules towards the perinuclear region during virus cell entry enhances the chance to productively infect a cell. Previous studies described two types of early endosomes that differ in their motility and maturation kinetics [15]. Rapidly maturing early endosomes are predominantly observed in the perinuclear region of the cell and the delivery of cargo to these organelles was found microtubule-dependent [15]. During endosomal maturation the composition of the endosome alters

and it is therefore possible that the rapidly maturing early endosomes in the perinuclear region have a more favourable lipid/protein composition for CHIKV fusion pore formation and thus genome delivery. This is strengthened by the observation that in presence of nocodazole viral hemifusion is restricted to the cell periphery and at these conditions gRNA delivery is impaired.

Transport towards rapidly maturing early endosomes may also promote the release of the nucleocapsid into the cell cytosol after fusion pore formation. Within the particle, the capsid protein is known to interact with the cytosolic domain of E2, however it is not known how this interaction is broken to release the viral nucleocapsid after membrane fusion. Recent data indicates that the CHIKV 6K protein can function as an ion channel [30] similar to what has been reported before for the M2 protein of Influenza virus (IAV) [31,32]. The M2 viroporin is essential for the weakening of the interactions between the IAV genome and its capsid via the influx of protons and potassium ions during endocytosis, thereby making the IAV capsid uncoating competent [32]. The mechanisms of CHIKV nucleocapsid release are poorly understood and further research is warranted to investigate if indeed rapidly maturing endosomes harbour an environment that is beneficial in this process.

An alternative explanation to our findings is that microtubules act at post-fusion conditions. In case of IAV, the release of the viral capsid in the cell cytosol was found to be dependent on the host protein histone deacetylase 6 (HDAC6) [33]. Upon IAV membrane fusion, the capsid is exposed at the cytoplasmic side of the endosome and recruits HDAC6 via unanchored ubiquitin chains in the IAV capsid. HDAC6 then serves as a linker molecule to the microtubule motors dynein and dynactin and the actin motor myosin II. While linked to the IAV capsid, these molecular motors are thought to generate opposing forces thereby breaking apart the IAV capsid resulting in the dispersal of the capsid and IAV gRNA in the cell cytosol. Also for HIV, interactions of the capsid with dynein and kinesin adaptor proteins have been shown to be important for nucleocapsid uncoating [13,19,34]. Bernard and co-workers suggested that microtubules are also important in CHIKV infection at post-fusion conditions [28]. The authors observed that nocodazole inhibited infection when CHIKV fusion and infection was artificially induced at the plasma membrane of cells. Although this is a commonly used strategy, the efficiency of fusion/infection is typically much lower under these experimental setting making it difficult to compare to natural infection conditions.

In conclusion, this study enhances our understanding on the interactions of CHIKV with the cytoskeleton by identifying and visualizing the complex trafficking patterns during CHIKV entry. Based on our data, we conclude that microtubules are required for efficient CHIKV genome delivery into the cytosol. Further studies are required to pinpoint via which mechanism microtubules affect the delivery of the CHIKV genome.

## Supporting information

**S1 Fig. Nocodazole treatment disrupts the microtubule network in U-2 OS.** (A) Representative pictures of the α-tubulin staining in U-2 OS cells in absence (left) and presence (right) of 10 μM nocodazole for 2 h. Scale bar: 25 μm. (B) Cell viability of U-2 OS cells upon treatment with nocodazole. U-2 OS cells were treated for 18 h, after which cell viability was assessed using standard ATPlite assay. Dotted line indicates 75% cell survival. A total of three experiments were performed in triplicate. Each dot represents the average of an independent experiment. Error bars represent SD.
(TIF)

**S2 Fig. Nocodazole inhibits CHIKV infection.** U-2 OS cells were pretreated for 2 h with 10 μM nocodazole and infected with CHIKV LR2006-OPY1 (A) or CHIKV LS3-GFP (B) at

MOI 1 or 20 for 10 h. (A) Flow cytometry analysis of E2-positive cells in the presence or absence of nocodazole. (B) Flow cytometry analysis of GFP-positive cells in the presence or absence of nocodazole. Data represents three independent experiments performed in triplicate, each dot represents the average of an individual experiment; error bar represents SD. Data was analyzed and compared to the control using student T-test ($^*p \leq 0.05$ and $^{****}p \leq 0.0001$); ns, not significant.
(TIF)

**S3 Fig. Nocodazole treatment disrupts the microtubule network in BS-C-1 cells.** (A) Representative pictures of the α-tubulin staining in BS-C-1 cells in absence (left) and presence (right) of 10 μM nocodazole for 2 h. Scale bar: 25 μm. (B) Cell viability of BS-C-1 cells upon treatment with 10μM nocodazole. The cells were treated for 12h, after which cell viability was assessed using standard ATP lite assay. Dotted line indicates 75% cell survival. Three independent experiments were performed in triplicate; each dot represents the average of a single experiment; error bar represents SD.
(TIF)

**S4 Fig. Nocodazole inhibits CHIKV infection in BS-C-1 cells.** BS-C-1 cells were pretreated for 2 h with 10 μM nocodazole and infected with CHIKV LS3-GFP at MOI 20 for 10 h. (A) Flow cytometry analysis of GFP-positive cells in the presence or absence of nocodazole. (B) Mean fluorescence intensity (MFI) of the infected population. Data is normalized to the positive control. Three independent experiments were performed in triplicate, each dot represents the average of an independent experiment; error bar represents SD. Data was analyzed and compared to the control using student T-test ($^*p \leq 0.05$).
(TIF)

**S5 Fig. Quantification of GAPDH and Rab5 in membrane and cytosolic fractions.** U-2 OS cells were permeabilized using 50μg/ml digitonin for 5 min at RT and subsequently 30 min on ice. Subsequently, the total amount of GAPDH and Rab5 was determined in the cytoplasmic and the membrane fraction by Western blot quantification. Four independent experiments were performed in duplicate, each dot represents the average of an independent experiment; error bar represents SD.
(TIF)

**S1 Movie. CHIKV trajectory showing fast-directed movement.** Movie showing a trajectory of a single virus particle displaying fast-directed movement before hemifusion. The trajectory is depicted color-coded, with purple and red representing the start and end of the trajectory, respectively. Recording was performed at 1 frame/s. Playback time is 15 frames/s. Virtual time and the color code for time are shown in the right down corner.
(AVI)

**S2 Movie. Trajectory of a CHIKV particle remaining relatively static during entry.** The trajectory is recorded and depicted as S1 Movie. Playback time is 30 frames/s.
(AVI)

**S3 Movie. CHIKV trajectory in nocodazole-treated cells.** The trajectory is recorded and depicted as S1 Movie. Playback time is 15 frames/s.
(AVI)

**S4 Movie. CHIKV trajectory in nocodazole-treated cells.** The trajectory is recorded and depicted as S1 Movie. Playback time is 10 frames/s.
(AVI)

## Author Contributions

**Conceptualization:** Tabitha E. Hoornweg, Izabela A. Rodenhuis-Zybert, Jolanda M. Smit.

**Formal analysis:** Tabitha E. Hoornweg, Ellen M. Bouma.

**Funding acquisition:** Jolanda M. Smit.

**Investigation:** Tabitha E. Hoornweg, Ellen M. Bouma, Denise P.I. van de Pol.

**Methodology:** Tabitha E. Hoornweg, Ellen M. Bouma, Jolanda M. Smit.

**Project administration:** Tabitha E. Hoornweg, Ellen M. Bouma, Izabela A. Rodenhuis-Zybert, Jolanda M. Smit.

**Resources:** Izabela A. Rodenhuis-Zybert, Jolanda M. Smit.

**Supervision:** Izabela A. Rodenhuis-Zybert, Jolanda M. Smit.

**Validation:** Izabela A. Rodenhuis-Zybert, Jolanda M. Smit.

**Visualization:** Tabitha E. Hoornweg, Ellen M. Bouma.

**Writing – original draft:** Tabitha E. Hoornweg, Ellen M. Bouma.

**Writing – review & editing:** Tabitha E. Hoornweg, Ellen M. Bouma, Denise P.I. van de Pol, Izabela A. Rodenhuis-Zybert, Jolanda M. Smit.

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
