## [Decision Letter · Decision Letter 0]

1 Apr 2020

Dear Dr. Smit,

Thank you very much for submitting your manuscript "Chikungunya virus requires an intact microtubule network for efficient viral genome delivery" for consideration at PLOS Neglected Tropical Diseases. As with all papers reviewed by the journal, your manuscript was reviewed by members of the editorial board and by several independent reviewers. In light of the reviews (below this email), we would like to invite the resubmission of a significantly-revised version that takes into account the reviewers' comments. 

We cannot make any decision about publication until we have seen the revised manuscript and your response to the reviewers' comments. Your revised manuscript is also likely to be sent to reviewers for further evaluation.

Sincerely,

Abdallah Samy

Deputy Editor

Reviewer's Responses to Questions

**Key Review Criteria Required for Acceptance?**

**Methods**

-Are the objectives of the study clearly articulated with a clear testable hypothesis stated?

-Is the study design appropriate to address the stated objectives?

-Is the population clearly described and appropriate for the hypothesis being tested?

-Is the sample size sufficient to ensure adequate power to address the hypothesis being tested?

-Were correct statistical analysis used to support conclusions?

-Are there concerns about ethical or regulatory requirements being met?

Reviewer #1: Yes to most questions, other than what's included in my comments below. 

No ethical/regulatory concerns.

Reviewer #2: Appropriate. It would have been useful to include an indication of the particle to infectious unit ratio, especially for the DiD labelled virus.

**Results**

-Does the analysis presented match the analysis plan?

-Are the results clearly and completely presented?

-Are the figures (Tables, Images) of sufficient quality for clarity?

Reviewer #1: Please see specific comments below

Reviewer #2: The results are for the most part clearly presented. I have just a few questions/comments.

Firstly, the authors show that cell viability is unaffected after 18hrs of nocodazole treatment (Fig S1B), but presumably cell division is inhibited and there are fewer cells in these cultures? If this is the case, does it impact on the data presented in S2B - if there are fewer cells in the nocodazole treated samples, presenting the data as the percentage of infected cells will underestimate the nocodazole inhibition? What are the dotted lines in S1B and S3B?

Why have authors chosen to assess the effects of nocodazole at 10 hrs post infection?

Using DiD as a fusion reporter does detect hemifusion, but it also reports full fusion. In these assays the authors cannot distinguish between hemifusion and full fusion. So I think it misleading to use hemifusion throughout the paper, and I suggest they use 'hemifusion/full fusion' instead.

Can the authors be certain that the fast directed movement described in Fig 2A is intracellular and not on the cell surface? I.e. could it be viral surfing?

Line 291 - there does seem to be a 30% change in MFI.

Figs 2A and B - the labeling of the boxes is defective.

Line 375 - how do the authors define perinuclear? The two events illustrated in Fig 2A do not appear to be perinuclear, yet in Fig 3/Line 398 the authors give a quantitative analysis of perinuclear v non-perinuclear fusion.

**Conclusions**

-Are the conclusions supported by the data presented?

-Are the limitations of analysis clearly described?

-Do the authors discuss how these data can be helpful to advance our understanding of the topic under study?

-Is public health relevance addressed?

Reviewer #1: Yes to all ?s.

Reviewer #2: The conclusions are appropriate for the data presented. However, I am concerned that the authors do not cite and discuss relevant published work appropriately. Specifically, papers by Vonderheit and Helenius (PLoS Biol 2005 e233) describing morphological analysis of Semliki Forest virus is not mentioned; and though cited, Bernard et al's. report of clathrin-independent internalization of CHIKV into cells is not discussed.

**Editorial and Data Presentation Modifications?**

Reviewer #1: Please see specific comments below

Reviewer #2: The paper should be carefully edited for word usage, etc.

**Summary and General Comments**

Reviewer #1: In this manuscript, Hoornweg et al elucidate the role of the microtubule network in the trafficking of CHIKV particles early in the course of infection. They hypothesize that microtubules may direct the particle to a cellular location that is beneficial for establishing infection or aids in nucleocapsid uncoating. They monitored trafficking of DiD-labelled viral particles and observed two distinct patterns prior to membrane fusion in the early endosomes. Using nocodazole to disrupt the microtubule network, they observed reduced number of infected cells, restriction of fusion to the cell periphery, and impaired delivery of the viral genome into the cytoplasm. 

The microtubule network has been previously implicated in the entry of multiple other viruses, and thus the innovation here is limited to the demonstration of this requirement for CHIKV and the discovery of the two distinct trafficking patterns. While the data demonstrating the two distinct viral populations is interesting, it remains unclear what is the functional relevance of this difference and what is the mechanistic role mediated by microtubules in CHIKV entry. 

Major comments:

1. The magnitude of the effect shown in most figures is small (less than a 2 fold difference in many cases – e.g. Fig. 1A, Fig. 4B), which makes it unclear what the biological relevance of the findings is. 

2. The number of infected cells is very low (2.9% in an MOI of 1, 9.2% in an MOI of 20 as per line 288). Why did the authors choose to infect the cells only for 30 min (when a more standard infection time is 1 hour for alphaviruses)? Perhaps a longer infection time will increase infection rate and improve the dynamic range?

3. The authors should demonstrate a dose-response effect for the various phenotypes shown with nocodazole treatment. 

4. Is it possible the cell cycle arrest at G2/M induced by nocodazole accounts for the observed reduction in infection and trafficking of viral particles?

5. The distance and velocity traveled by individual viral particles needs to be shown in figures 2 and 3. 

6. In Fig. 4A, the Western blot should include all 3 conditions (control, and two treatments). 

Additional comments: 

1. Graphs in all figures – would be better to show controls in each figure rather than showing just the % (or alike) difference relative to control.

2. Would be good to avoid single column graphs (as in 1D). These can be just mention in the text.

3. Figure 1E – a schematic showing time of drug addition would be helpful.

4. Figure 3D – the legend should explain what before, during and after means. 

5. Embedding figure legends in the text makes it harder to review.

Reviewer #2: Overall, this work provides a modest step forward in understanding CHIKV entry into cells. Unfortunately, there is really no indication of how microtubule based transport of CHIKV favors penetration and infection.

PLOS authors have the option to publish the peer review history of their article (what does this mean?). If published, this will include your full peer review and any attached files.

Reviewer #1: No

Reviewer #2: No
---

## [Decision Letter · Decision Letter 1]

10 Jun 2020

Dear Dr. Smit,

We are pleased to inform you that your manuscript 'Chikungunya virus requires an intact microtubule network for efficient viral genome delivery' has been provisionally accepted for publication in PLOS Neglected Tropical Diseases.

Best regards,

Abdallah M. Samy, PhD

Deputy Editor

---

## [Editor Report · Acceptance letter]

14 Jul 2020

Dear Dr. Smit,

We are delighted to inform you that your manuscript, "Chikungunya virus requires an intact microtubule network for efficient viral genome delivery," has been formally accepted for publication in PLOS Neglected Tropical Diseases.

Best regards,

Shaden Kamhawi

co-Editor-in-Chief

Paul Brindley

co-Editor-in-Chief
